# Adsorption and Photodegradation of Lanasol Yellow 4G in Aqueous Solution by Natural Zeolite Treated by CO_2_-Laser Radiation

**DOI:** 10.3390/ma16134855

**Published:** 2023-07-06

**Authors:** David Correa-Coyac, Alexandre Michtchenko, Gregorio Zacahua-Tlacuatl, Yair Cruz-Narváez, José J. Castro-Arellano, Monserrat Sanpedro-Díaz, Carlos F. de J. Rivera-Talamantes, Yury M. Shulga

**Affiliations:** 1Instituto Politécnico Nacional, SEPI-ESIME-Zacatenco, Av. IPN S/N, Ed.5, 3-r piso, Ciudad de México 07738, Mexico; dcoyac@outlook.com (D.C.-C.); carlosrivera182mx@hotmail.com (C.F.d.J.R.-T.); 2Laboratorio de Posgrado e Investigación de Operaciones Unitarias-ESIQIE, Instituto Politécnico Nacional, Zacatenco, UPALM, Zacatenco, Col. Lindavista, Ciudad de México 07738, Mexico; yair_8_88@hotmail.com (Y.C.-N.); jjcastro@ipn.mx (J.J.C.-A.); monserrat.sanpedro.20@gmail.com (M.S.-D.); 3Federal Research Center of Problem of Chemical Physics and Medicinal Chemistry, Russian Academy of Sciences, Moscow 142432, Russia; yshulga@gmail.com

**Keywords:** natural zeolite, laser radiation, Lanasol Yellow 4G, adsorption, advanced oxidation process, photo-Fenton

## Abstract

Natural zeolite is a widely used material with excellent environmental cleaning performance, especially in water and wastewater treatment. Natural zeolite (Z_ini_) calcined by CO_2_-laser radiation (ZL) was tested as a catalyst for the photodegradation and the adsorption of industrial azo dye Lanasol Yellow 4G (LY4G) in water. Morphology, chemical structure, and surface composition of Z_ini_ and ZL were analyzed by XRD, SEM, EDS, and XPS. UV/Visible spectrophotometry was used to evaluate the photocatalytic activity of Z_ini_ and ZL. The photocatalytic activity of the studied zeolites was associated with the presence of Fe oxides in their composition. Laser-treated natural zeolite showed higher efficiency as a photocatalyst compared to untreated natural zeolite.

## 1. Introduction

The textile and clothing industries satisfy fundamental human needs and employ millions of people worldwide [1,2]. However, textile companies are responsible for producing several industrial pollutants, contributing to ecological problems, especially waterbody pollution [3,4]. Untreated effluents discharged into bodies of water contain mainly a mixture of dyes, metals, and other pollutants [5]. Even at low concentrations, dyes are undesirable in aquatic environments because they limit light penetration and interfere with the growth of aquatic plants. The amount of oxygen in ecosystems decreases due to a light reduction [6,7,8].

Azo dyes are the most popular type of synthetic dyes. All azo dyes necessarily contain one or more azo groups (–N=N–). Figure 1 shows the molecular structure of one of the azo dyes, namely Lanasol Yellow 4G. The azo group is responsible for the color of azo dyes, linking two aromatic groups together [9]. The presence of these azo groups in certain dyes raises concerns about potential human health risks and environmental pollution [10]. Consequently, it is essential to follow proper safety and disposal procedures when using these dyes, as they can be hazardous to human health and the environment if not handled correctly. Therefore, they must be removed from the wastewater to fulfill effluent regulations before disposal [11,12,13,14]. To comply with strict environmental regulations, many techniques have been adopted to remove dyes from water and wastewater [15,16,17,18,19,20,21,22,23]. Several methods have been documented for the removal of Lanasol Yellow 4G from aqueous solutions. Gallinaceous feathers have been shown as a unique adsorbent for the LY4G azo dye [24]. Also, the impact of oxidants on the photoelectrocatalytic decolorization of LY4G has been examined using a nanocomposite of α-Fe_2_O_3_, TiO_2_, and activated charcoal under visible light conditions [25]. Additionally, sonocatalytic degradation has been studied as a viable approach, with a specific focus on the use of synthesized ZrO_2_ nanoparticles on biochar for the degradation of this dye [26]. In this study, we evaluate the potential of natural zeolite and natural zeolite calcined by CO_2_-laser radiation, both as an adsorbent and as a photocatalyst for dye removal, with the aim of proposing a novel and effective treatment method.

Advanced oxidation processes (AOPs) are among the best and most promising technologies for the reduction of toxicity and completely mineralizing recalcitrant contaminants of wastewater, such as aromatics, pesticides, and synthetic organic dyes [27,28]. Various advanced oxidation processes for dye degradation have been developed in recent years [29,30,31,32,33]. The application of AOPs essentially involves the production of reactive species, such as hydroxyl radicals (•OH), sulfate radicals (SO_4_•^−^), and superoxide radicals (O_2_^−^•)), that contribute to the degradation of organic pollutants [16,28,34,35].

AOPs refer to a set of oxidative water treatments that may be classified into two main groups: procedures that include the use of radiation and techniques that do not involve the use of radiation. Radiation-assisted AOPs processes include photolysis, photocatalysis, and the photo-Fenton process, while non-irradiation processes include the Fenton-based process, electrochemical oxidation, electrical discharge, ozonation, and sonolysis, among others [23,36,37].

According to the literature, the photo-Fenton reaction under acidic conditions (about pH 2.8) offers high levels of degradation due to high solubility and good stability of Fe(III), which increases the rate of HO• radical formation [38,39,40]. The goal of this study is to determine the optimal parameters for increasing the removal of azo dye in a photo-Fenton process that involves Fe(III) at a pH of 2.5 in different aqueous matrices. Both natural and synthetic zeolites can be beneficial materials for a variety of industries as adsorbents, soil modifiers, and ion exchangers [23,41].

Natural zeolites are abundant and cheap materials that have been widely used as adsorbents for dyes and various other pollutants during water and wastewater treatment processes [42,43,44]. They also have demonstrated significant catalytic activity in the oxidation of azo dyes [31,45,46,47,48]. They are interesting candidates as heterogenous catalysts for wastewater treatment by Fenton and photo-Fenton oxidation. The Fenton process is a widely used method for wastewater treatment due to its rapid Fe/H_2_O_2_ reactions, relatively cheap input chemicals, and the ability to degrade many organic materials [49,50].

In this study, the potential for photodegradation of the azo dye Lanasol Yellow 4G (LY4G) in water was investigated using an inexpensive iron-containing natural zeolite from the Catano-Etla region of Oaxaca State, Mexico. In a novel approach, CO_2_-laser radiation was employed to excite the zeolite and potentially enhance its catalytic activity. As reported in the literature, laser radiation treatment can result in significant modification of the catalyst surface in certain cases [51]. Laser irradiation was chosen as a calcination technique for this study due to its precision in controlling exposure duration and energy, allowing for finely tuned adjustments to the calcination process. This method guarantees a rapid, uniform heating rate, resulting in the production of high-quality, homogeneous photocatalysts while minimizing any non-uniformity that is frequently found with conventional heating techniques. Furthermore, the natural zeolite and the laser-irradiated zeolite samples were evaluated for their efficacy as adsorbents for the mentioned dye.

## 2. Materials and Methods

### 2.1. Chemical and Reagents

Natural zeolite from the region of Etla, Oaxaca State, Mexico was ground using a steel ball mill. The mill shell had an internal diameter and length of 0.2032 m. The grinding process was conducted for a duration of 3 h. A total of 80 balls ranging in diameter from 1 cm to 4 cm were utilized in the grinding process. Ferric chloride (FeCl_3_), and hydrochloric acid (HCl, 36%) were provided by Meyer, Mexico City, Mexico, while Hydrogen peroxide (H_2_O_2_, 50%) was provided by Alquimia, Mexico City, Mexico. All the aqueous solutions were prepared with Milli-Q water. Lanasol Yellow 4G (C_19_H_12_BrCl_2_N_5_Na_2_O_8_S_2_) was supplied by Novartis, Mexico City, Mexico. The chemical structure of LY4G is shown in Figure 1. The industrial dye was used without further purification. The solution was irradiated using a 13 W UV-A lamp, with λ_max_ = 355 nm, provided by Lumiaction^®^, Taipei, Taiwan.

### 2.2. Laser Radiation Treatment

Natural zeolite powder was thermally treated using a continuous wave CO_2_-laser (λ = 10.6 µm) with a maximum power output of 75 Watts. Therefore, the laser radiation treatments involved a three-stage temperature profile. The initial stage of the laser radiation treatment involves a preheating period, which is essential for elevating the temperature of the samples close to that of the calcination process. Five seconds of preheating was utilized in every laser radiation treatment.

The second stage of the laser radiation treatment entails the calcination of the catalyst, carried out over varying durations (15, 30, 60, and 240 s) and across a range of temperatures (300, 550, and 700 °C). Finally, the third stage encompasses the spontaneous decline in temperature that occurs after the laser is switched off, marking the conclusion of the laser calcination period.

To facilitate clarity, ZL samples have been denoted by two indices, time, and temperature. For instance, ZL_T_^t^ represents the natural zeolite calcined by CO_2_-laser radiation (ZL), where t is the time and T is the temperature of laser calcination. For example, the identifier ZL_550_^15^ indicates that the sample was annealed at 550 °C for 15 s. The natural zeolite sample, already introduced as Z_ini_, serves as the reference point.

### 2.3. Characterization of Photocatalyst

The surface area of Z_ini_ and ZL was analyzed using the Brunauer-Emmett-Teller (BET) technique, and the pore size distribution was determined by the Barrett-Joyner-Halenda (BJH) method, employing a Micromeritics ASAP 2045N instrument (Norcross, GA, USA). The crystal structure was obtained using an X-ray diffractometer (XRD, SIEMENS D500, Erlangen, Germany) with an operating voltage of 35 kV. The surface morphology and chemical composition of the samples were examined with a scanning electron microscope (SEM, JEOL JSM-7800F, Tokyo, Japan) coupled with energy-dispersive X-ray spectroscopy (EDS). Surface chemical composition was also investigated using X-ray photoelectron spectroscopy (XPS, Thermo Fisher K-Alpha, Waltham, MA, USA). Finally, a UV-Vis spectrophotometer (Nanodrop One C, Thermo Fisher Scientific, Waltham, MA, USA) was employed to evaluate the degradation efficiency of the azo dye LY4G.

### 2.4. Adsorption Experiments

Adsorption experiments were carried out in quintuplicate, and average results were calculated to ensure the reliability and reproducibility of the data. The adsorption studies of Z_ini_ and ZL_550_^15^ were conducted in 40 mL glass flat-bottom vials under a constant stir rate of 180 rpm to evaluate their efficiency in removing the LY4G dye from aqueous solutions. The adsorption experiments were designed to evaluate the effects of various parameters including pH, and zeolite type (either Z_ini_ or ZL_550_^15^), on the discoloration performance of the LY4G dye. Adsorption experiments were conducted at a controlled environment of 24 °C in the dark. A solution of 35 mL of the LY4G dye, at a concentration of 150 ppm, was created using high-purity Milli-Q water. Then, 2 mg of zeolite was integrated into the solution, initiating the adsorption procedure.

### 2.5. Photodegradation Experiments

The photocatalytic activities of Z_ini_, ZL_550_^15^, ZL_550_^30^, ZL_550_^60^, and ZL_550_^240^ were evaluated using the photo-Fenton process in a Pyrex glass reactor. This reactor, with a volume of 500 mL, a diameter (di) of 70.7 mm, and a depth (h) of 155 mm, was stirred at a rate of 700 rpm. Each experiment utilized a reaction mixture comprising 20 mg of the specified zeolite and 28 mg of FeCl_3_ dispersed in a 350 mL solution with a concentration of 150 ppm of LY4G dye. This resulted in a water layer with a thickness of approximately 89 mm. Then the pH of the solution was adjusted to 2.5 (using the proper HCl solution), and 4 mL of hydrogen peroxide was then added. The work pressure was atmospheric pressure (585 mm Hg). Experimental tests were conducted using an average UV-A light intensity of 360 W/m^2^ (Appendix A). The UV-A lamp was submerged in the solution, and the exterior of the reactor was covered with aluminum foil to prevent light leakage. The reactor also incorporates a water-cooling jacket to dissipate heat generated by the lamp, while keeping the temperature steady. The cooling water temperature was set to 24 °C with a flow of 13 L/min for all tests. Figure 2 illustrates the reactor assembly used in the photocatalytic tests.

### 2.6. Evaluation of Degradation Efficiency and Adsorption Capacity

The evaluation of azo dye concentration was conducted using UV/Vis spectrophotometry. Measurements were taken every 30 min for an hour in the adsorption study, and every 5 min over 30 min in the photodegradation test. Samples were centrifuged before concentration measurements were taken. The UV/Vis spectra of LY4G were recorded from 200 to 700 nm, with the maximum absorbance wavelength found at 400 nm. The absorption intensity at 400 nm can be used to calculate the concentration of LY4G at specific time intervals in adsorption and photocatalytic studies. The degradation efficiency (%) was calculated using Equation (1):(1)$$degradation\;efficiency\left(\%\right)=100\%\times \frac{\left(C_{0}-C\right)}{C_{0}}$$
where *C*_0_ and *C* represent the initial and final concentrations of the azo dye, respectively. Furthermore, Equation (2) was used to determine the adsorption capacity (*Q*, mg g^−1^).
(2)$$Q=\frac{\left(C_{i}-C_{e}\right)V}{W}$$
where *C_i_* and *C_e_* are the initial and the equilibrium concentration of the adsorbate (mg L^−1^) respectively, *V* is the volume of the solution containing the adsorbate (L) and *W* is the mass of the adsorbent (g).

## 3. Results

### 3.1. Surface Area and Porosity Analysis

The calcination temperature of natural zeolites is a crucial factor in the activation process, which influences the surface area of these materials [52,53,54,55]. According to the literature, calcination temperature significantly affects the number of active sites, crystallinity, and pore size distribution. Consequently, comprehending and manipulating calcination conditions emerges as an indispensable aspect in the pursuit of harnessing the full potential of natural zeolites across diverse applications [56,57].

In this study, Brunauer-Emmett-Teller (BET) analyses were conducted on Z_ini_ and ZL samples at calcination temperatures of 300, 550, and 700 °C for a duration of 15 s, (ZL_300_^15^, ZL_550_^15^, and ZL_700_^15^, respectively). Table 1 details the change in surface area and porosity properties caused by different temperature calcination processes applied to natural zeolites using CO_2_-laser radiation.

The samples exposed to different calcination temperatures demonstrate significant variations in surface area, pore volume, and average pore radius. Among these temperatures, calcination at 550 °C proves to be the most effective method in enhancing the surface and pore volume areas (BET and BJH) of the ZL samples. In contrast, the calcination process carried out at 300 °C and 700 °C leads to a decrease in surface area in contrast to the original Z_ini_ sample. Interestingly, the pore radius values exhibit a general decline as the calcination temperature increases, with the smallest pore radius corresponding to the ZL_550_^15^ sample. It has been observed that an increase in surface area often correlates with an enhancement in pore volume and a reduction in pore radius. This correlation could potentially lead to the formation of bottleneck-shaped pores, signifying a noteworthy morphological transformation in the material [58]. A comprehensive literature review reveals a lack of studies specifically addressing the relationship between pore size and its impact on adsorption capacity and photocatalytic activity in natural zeolites. Nonetheless, the physical state of occluded molecules in zeolites depends on the size and structure of the pores in the host material [59,60]. Based on prior studies, it has been established that calcination treatment is the most effective method for modifying the properties of natural zeolite. According to the literature data, a temperature of 550 °C is defined as the optimal temperature for annealing natural zeolites [30,45,61]. Considering these findings, the XRD characteristics of Z_ini_ and ZL_550_^15^ samples are further explored.

### 3.2. X-ray Diffraction Analysis

Figure 3 illustrates the X-ray diffraction patterns of Z_ini_ and ZL_550_^15^ samples, obtained using a 2*θ* range of 5 to 60°, with a step of 0.02° and a scanning velocity of 2°/min. The phases were identified through peak comparison with JCPDS files including clinoptilolite (025-1349), erionite (022-0854), mordenite (06-0239), heulandite (21-131), feldspar (025-0618), and quartz (033-1161). The major mineralogical phase identified through XRD analysis was a clinoptilolite-heulandite type zeolite, displaying typical patterns. 

Table 2 presents the estimates of the degree of crystallinity (*DC*) and crystallite size (D) for the 5 samples studied. The evaluation of the *DC* was carried out according to Equation (3).
(3)$$DC=\frac{\int_{5}^{60}I\left(2\theta \right)d\theta -\int_{5}^{60}F\left(2\theta \right)d\theta }{\int_{5}^{60}I\left(2\theta \right)d\theta }$$
where *I* is the intensity of the X-ray pattern, and *F* is the intensity of the X-ray background (see Figure 3). The crystallinity of the studied samples changes slightly. Crystallite size estimation was performed using the Scherrer formula on the three narrowest peaks located at approximately 9.7°, 25.5°, and 41.0°.

It is worth noting that the sizes of crystallites differ to some extent along different crystallographic directions. Specifically, for the crystallographic direction corresponding to 2*θ* = 41.0°, the crystallite size remains practically unchanged (within the range of 28.9–29.9 nm) with respect to annealing temperature. Although more noticeable changes are observed for other crystallographic directions, clear calcination times dependencies are not evident. As can be seen from Table 2 after CO_2_-laser calcination in the zeolite, the degree of crystallinity and the size of the crystallites somewhat change.

### 3.3. Morphology and Chemical Composition of Zeolite

Based on the analysis of the surface area and porosity, ZL_550_^15^ exhibited higher values following CO_2_ laser radiation treatment. Consequently, ZL_550_^15^, along with Z_ini_, was chosen for SEM analysis.

The SEM images for the Z_ini_ and ZL_550_^15^ samples at various magnifications are depicted in Figure 4. The comparison between the SEM micrographs of Z_ini_ sample (Figure 4a,b) and ZL_550_^15^ (Figure 4c,d) reveals notable morphological differences and shifts in particle size distribution due to the calcination process by CO_2_-laser radiation.

Specifically, the SEM micrographs of ZL_550_^15^, captured at 500× and 25,000× magnifications, demonstrate the presence of notably smaller particles and decreased particle agglomeration in comparison to the Z_ini_ sample. This alteration in morphology and particle size distribution not only reflects the impact of the calcination process, but also potentially influences the adsorption and photodegradation capabilities of the zeolite, which is a significant factor in the context of our study.

The EDS data presented in Table 3 indicates the presence of expected elements such as silicon, aluminum, oxygen, and carbon, as well as magnesium, sodium, potassium, calcium, and iron in the analyzed samples. Iron is well-known for its exceptional performance as a photocatalyst in dye photodegradation processes. According to the data presented in Table 3, it can be observed that the Si/Al ratio for Z_ini_ is 4.67, while for ZL_550_^15^, it is 4.05. This suggest that the annealing process by laser radiation at 550 °C may induce certain alterations in the sample composition in the analyzed region.

### 3.4. XPS

Considering the findings from previous analyses of the sample treated at 550 °C, we proposed to modify the duration of the laser radiation treatment. The objective was to investigate how this variable affects the elemental content in the surface layer of the samples.

The study of the surface composition of natural zeolite and the impact of annealing on its near-surface layer composition was further pursued using the XPS method. An overview spectrum of the original sample is shown in Figure 5. Upon close examination of the sample under study, it has been determined that the surface contains not only the elements previously detected through the EDS method, but also traces of fluorine and nitrogen. According to the data presented in Table 4 referring to the composition of the samples under study, it is evident that the use of laser radiation treatment results in a diminution in the concentration of oxygen in the near-surface region. This decrease can be attributed to the partial restoration of the surface. Together with a decrease in oxygen concentration, an increase in carbon concentration occurs. In some cases, excessive carburization can lead to gasification of the support, which can further impact the activity of catalyst. Therefore, controlling the carburization conditions is crucial for maintaining the desired catalyst activity in various reactions [62,63].

### 3.5. Effect of pH on Dye Adsorption

The effect of pH on the adsorption of LY4G dye by Z_ini_ and ZL_550_^15^ samples was evaluated. In this study, the adsorption capacities Q (see Equation (2)) of these samples were determined at certain pH levels (2.5, 4, 5.5 and 7) to determine the optimal pH for maximum adsorption efficiency and to understand the effect of pH on the interaction between the adsorbate and adsorbent. The equilibrium concentration (C_e_) was measured at t = 60 min.

Based on the adsorption capacity values at different pH levels (Table 5), it can be concluded that the adsorption process for both Z_ini_ and ZL_550_^15^ samples is most favorable under acidic conditions, particularly at pH 2.5. For both samples, the adsorption capacity decreases as the pH increases, reaching its lowest value at the neutral pH of 7. This trend is consistent with previous studies conducted on natural zeolites from the same region [61]. Both samples demonstrated relatively lower adsorption capacities, with Z_ini_ achieving a removal efficiency of 6.2% and ZL_550_^15^ achieving 4.4%. This phenomenon could potentially explain the observed correlation with pH levels.

Under acidic conditions, the higher concentration of [H_3_O]^+^ ions in the solution can protonate the surface functional groups of the zeolite particles, resulting in a decrease in the effective surface charges of the material. The dissociation of the dye in an aqueous medium is favored at lower pH levels. Thus, the anion of the dye formed during dissociation has the potential to be adsorbed at the cationic sites of the zeolite. This phenomenon can provide an explanation for the observed pH dependence.

### 3.6. Evaluation of Photocatalytic Activity

In Figure 6a, the spectrum of LY4G solutions is presented across a range of concentrations. The LY4G spectrum exhibit a notable peak at 400 nm, indicating its highest level of absorbance within this range. A calibration curve for the azo dye is constructed by using the absorbance values at this specific wavelength for different concentrations, as demonstrated in Figure 6b. 

The catalytic efficiency of Z_ini_ and ZL zeolites, calcined at 550 °C (ZL_550_), in the bleaching reaction of an aqueous LY4G dye solution. The results are presented in Figure 7, where the comparative performance of four different samples of ZL_550_ samples. Each sample was subjected to different pre-annealing times (15, 30, 60, and 240 s). 

The results of the experiment, as shown in Figure 7, indicate that the ZL_550_^15^ sample, calcined for 15 s, demonstrated the most effective degradation efficiency over most reaction time intervals, while the original Z_ini_ sample exhibited the lowest efficiency in this regard. This can be attributed to the presence of Fe species detected in samples through EDS analysis. The Fe species are known to enhance the photocatalytic activity of zeolites. Thus, a higher presence of Fe species contributes to more degradation efficiency. In particular, the degradation efficiency of ZL_550_^15^ was at 71.9% within the first 5 min of the reaction, while the Z_ini_ sample was only at 42.7%. As the reaction progressed to 30 min, the degradation efficiency of ZL_550_^15^ peaked at 92%, while the Z_ini_ sample exhibited a degradation efficiency of 87.4%. These observations suggest that the calcination duration significantly affects the photocatalytic capabilities of the ZL samples, especially in the first minutes of the reaction.

## 4. Conclusions

In this study, the adsorption and photocatalytic properties of natural zeolite and natural zeolite calcined by CO_2_-laser radiation were evaluated for the removal of Lanasol Yellow 4G. The results revealed the influence of calcination temperature and duration on the degradation processes. It was observed that natural zeolite treated with CO_2_-laser radiation at 550 °C for 15 s exhibited the most significant enhancement of LY4G degradation, outperforming both the uncalcined natural zeolite and those subjected to alternative calcination conditions. The CO_2_-laser calcination treatment enhanced the surface area and pore volume of ZL_550_^15^ sample, simultaneously decreasing the pore radius. The degradation efficiency of ZL_550_^15^ reached 71.9% within just 5 min, surpassing the original Z_ini_ sample which achieved an efficiency of 42.7%. These data highlight the importance of CO_2_-laser radiation to optimize the catalytic efficiency of natural zeolite for LY4G degradation. Both samples demonstrated relatively lower adsorption capacities, with Z_ini_ achieving a removal efficiency of 6.2% and ZL_550_^15^ achieving 4.4%. However, despite these limitations, CO_2_-laser radiation treatment proved effective in amplifying the photocatalytic performance of natural zeolite for LY4G degradation. These findings highlight the potential of CO_2_-laser radiation to optimize the catalytic efficiency of natural zeolite, promising for real-world applications in wastewater treatment. 

## Figures and Tables

**Figure 1 materials-16-04855-f001:**
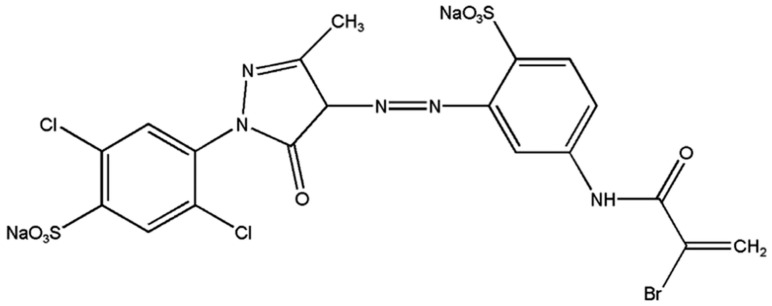
Molecular structure of Lanasol Yellow 4G. Adapted from the literature.

**Figure 2 materials-16-04855-f002:**
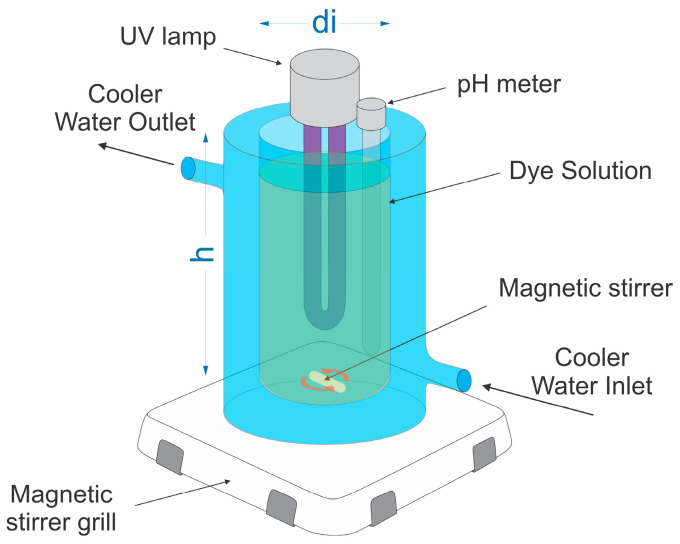
Schematic representation of the photocatalytic reactor assembly.

**Figure 3 materials-16-04855-f003:**
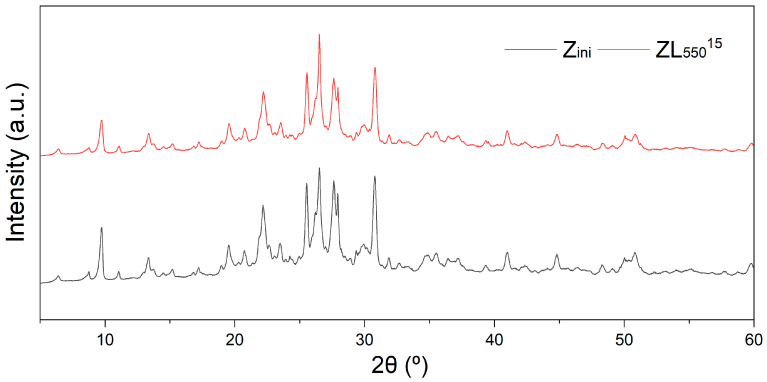
X-ray Diffraction patterns of natural zeolite (Z_ini_) and CO_2_-laser calcined zeolite (ZL_550_^15^).

**Figure 4 materials-16-04855-f004:**
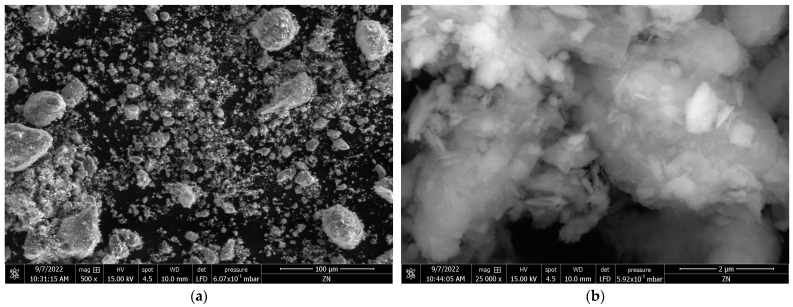
SEM micrographs of Z_ini_ (**a**,**b**) and ZL_550_^15^ (**c**,**d**) samples at 500× and 25,000× magnifications, respectively.

**Figure 5 materials-16-04855-f005:**
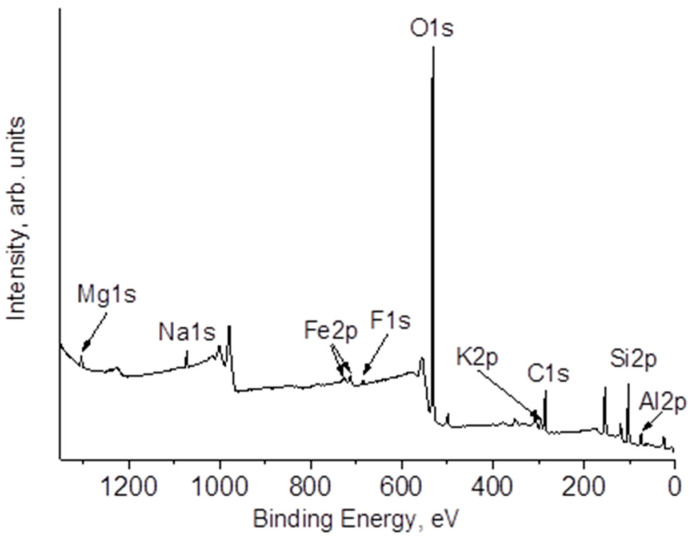
XPS survey spectrum of Z_ini_.

**Figure 6 materials-16-04855-f006:**
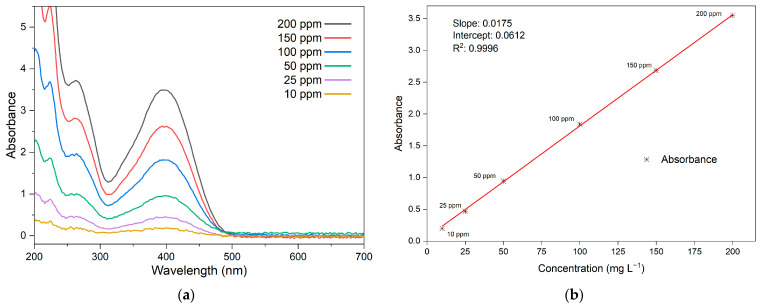
(**a**) Absorbance spectra of LY4G solutions at various concentrations and the corresponding (**b**) calibration curve at 400 nm.

**Figure 7 materials-16-04855-f007:**
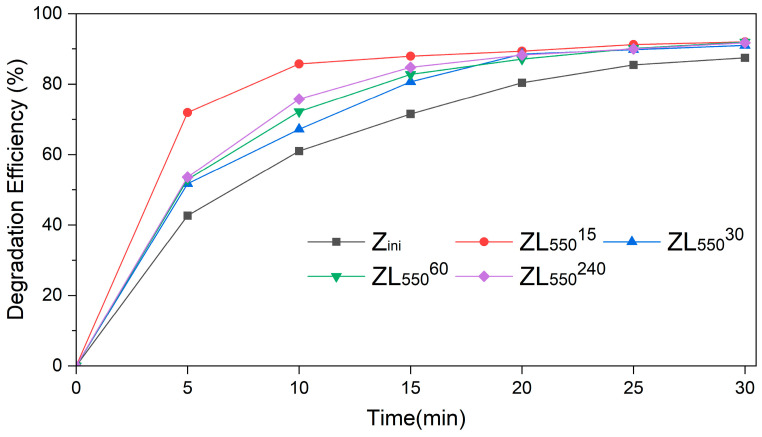
The degradation efficiency of Z_ini_ and ZL samples in the photo-Fenton reaction.

**Table 1 materials-16-04855-t001:** Surface area, pore volume, and pore radius for Z_ini_ and ZL samples calcined at different temperatures for 15 s.

Sample	Calcination Temperature (°C)	Surface Area (BET) (m^2^/g)	Pore Volume (×10^−2^ cm³/g)	Pore Radius DV (r) (Å)
Z_ini_	0	23.6	2.3	22.6
ZL_300_^15^	300	19.3	2.6	21.4
ZL_550_^15^	550	25.8	6.8	16.9
ZL_700_^15^	700	13.3	4.8	17.2

**Table 2 materials-16-04855-t002:** Crystallinity and grain size of Z_ini_, ZL_550_^15^, ZL_550_^30^, ZL_550_^60^, and ZL_550_^240^ catalysts.

Sample	Calcination Temperature (°C) and Time (s)	Crystallinity (%)	2*θ* (Degree)	D (nm)
Z_ini_	-	28.46	9.7	34.9
			25.5	42.2
			41.0	29.8
ZL_550_^15^	550; 15	28.50	9.7	29.7
			25.5	40.4
			41.0	28.9
ZL_550_^30^	550; 30	30.15	9.7	32.7
			25.5	42.9
			41.0	29.9
ZL_550_^60^	550; 60	24.34	9.7	28.7
			25.5	39.3
			41.0	29.8
ZL_550_^240^	550; 240	29.21	9.7	33.6
			25.5	41.1
			41.0	29.4

**Table 3 materials-16-04855-t003:** Elemental composition (%) and absolute error (%) of Z_ini_ and ZL_550_^15^ samples determined by EDS analysis.

Elements	Z_ini_		ZL_550_^15^	
	Mean %	Abs. Error %	Mean %	Abs. Error %
C	20.7	1.3	22.9	1.5
O	53.6	2.9	51.2	2.9
Al	3.9	0.2	4.4	0.2
Si	18.2	0.8	17.8	0.7
Na	1.2	0.1	1.0	0.1
Mg	0.20	0.03	0.20	0.02
K	0.90	0.04	1.00	0.04
Ca	0.80	0.03	0.80	0.03
Fe	0.50	0.04	0.80	0.04

**Table 4 materials-16-04855-t004:** The content of elements in the surface layer of the studied samples (in atomic %), determined from the integrated intensities of analytical lines in the XPS spectra.

Elements	Z_ini_	ZL_550_^15^	ZL_550_^30^	ZL_550_^60^	ZL_550_^240^
O	54.2	53.5	50.2	50.0	46.5
Si	19.1	18.9	17.7	18.0	16.1
Fe	1.0	1.0	1.0	0.9	0.9
C	14.4	15.6	21.5	21.2	27.5
Al	6.1	6.2	5.5	5.5	5.1
Mg	1.2	1.2	0.9	1.0	0.7
Na	1.1	1.2	1.0	1.1	1.0
Ca	0.8	0.8	0.7	0.7	0.7
F	0.8	0.7	0.6	0.6	0.5
K	0.7	0.6	0.6	0.6	0.5
N	0.6	0.4	0.4	0.4	0.5

**Table 5 materials-16-04855-t005:** Effect of pH on adsorption studies at equilibrium (60 min).

pH	Z_ini_ (Q, mg g^−1^)	ZL_550_^15^ (Q, mg g^−1^)
2.5	162.1	116.0
4	62.4	95.7
5.5	82.5	52.7
7	59.6	39.0

## Data Availability

The data presented in this study are available on request from the corresponding author. The data are not publicly available due to privacy.

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
