# Peer review of "Adsorption and Photodegradation of Lanasol Yellow 4G in Aqueous Solution by Natural Zeolite Treated by CO2-Laser Radiation"

_materials, 2023, doi:10.3390/ma16134855_

Round 1

Reviewer 1 Report

In this study, Natural zeolite (Zini) calcined by CO2-laser  radiation (ZL) was tested as a catalyst for the photodegradation of industrial azo dye Lanasol Yellow  4G (LY4G) in water. The morphology, chemical structure, and surface composition of Zini and one treated sample ZL550-15 were analyzed by XRD, SEM, EDS, and XPS. The effect of pH on Dye Adsorption was investigated on two zeolite samples Zini and ZL550-15 samples. The activity of Zini and ZL samples in the photo-Fenton reaction was tested.

To be published in the Journal of Materials, this manuscript needs major revision.

Please see the specific comments below:

In this study, Natural zeolite (Zini) calcined by CO2-laser  radiation (ZL) was tested as a catalyst for the photodegradation of industrial azo dye Lanasol Yellow  4G (LY4G) in water. The morphology, chemical structure, and surface composition of Zini and one treated sample ZL550-15 were analyzed by XRD, SEM, EDS, and XPS. The effect of pH on Dye Adsorption was investigated on two zeolite samples Zini and ZL550-15 samples. The activity of Zini and ZL samples in the photo-Fenton reaction was tested.

To be published in the Journal of Materials, this manuscript needs major revision.

Please see the specific comments below:

-          Introduction: An overview of adsorbents and methods used for Lanasol Yellow 4G in aqueous solution should be added.

-          The most severe limitations of the manuscript are physic-chemical properties and composition, excluding specific surface analysis, and the adsorption capacity of the material was only performed on two samples Zini and ZL55015, so the relationship between zeolite treatment modes, the properties and adsorption capacity as well as the photo-Fenton activity of the materials have not been elucidated, leading to low scientific soundness.

-          Experimental methods have not been fully described: The method and instrument for adsorption investigation must be described in more detail (lines 125-126, page 4); The capacity and dimensions of the photo-Fenton  process, as the thickness of the water layer, the magnetic stirrer, and the information of the A 13 W Ultraviolet A light (UV-A) such as lamp intensity, etc., should be added and specified (page 4)

-          The statement "These changes in peak intensities indicate modifications in the crystalline structure of the zeolite after CO2-laser calcination," (line 204) based on XRD analysis results "Upon calcination, the peak intensities of the diffractograms underwent noticeable changes, although no peaks appeared or disappeared. Specifically, the peaks at 9.74°, 200 22.18°, 25.54°, 27.62°, 27.92°, and 30.78° decreased in intensity, while the peak at 26.5° increased. For example, the peak at 9.74° declined from 22,353 (Zini) to 14,927 (ZL550-15), while the peak at 26.5° ascended from 42,984 (Zini) to 45,100 (ZL550-15)." (Lines 199-202) is not convinced. In addition, the crystallinity degree and the crystal size should be determined.

-          The issues that need to be explained and clarified:

+ Please explain why after Laser Radiation treatment pores radius of zeolite decreases; especially, sample ZL700-15 has a smaller pore radius, and larger pore volume, but the surface area (BET) is smaller than the sample treated at 300 oC and 550 oC? (Table 1.). And why “On the other hand, the calcination process carried out at 300°C and 700°C leads to a decrease in surface area in contrast to the original Zini sample.” (lines 177-178). Please explain why zeolite is not treated at 400, 450 oC, and 600 and 650 oC?

+ Please explain why “For both samples, the admissible capacity as the pH increases, with the lowest capacity observed at the neutral pH of 7” (253-254) and why the adsorption capacity (Q) of ZL550-15 is less than the Zini sample (table 4)

+ Is the quantity value ??????a???? ?????????y calculating according to Eq.1 included the amount of adsorption? So, the result in Figure 8 needs to be subtracted Q value at pH 2.5, so how much is left?

- Conclusions should be drawn from the results obtained. In this manuscript, some conclusions are not consistent with the results, such as

“The results show that factors such as calcination temperature and duration, and pH significantly influence the adsorption and degradation processes” (286) have not been proven enough, because the adsorption was only investigated on two samples Zini and Z55015, while degradation was only investigated at pH 2.5.

“This study also offers valuable information on the potential applications of natural zeolite-based materials as adsorbents”(lines 293-294) is not convincing because the Q value of the Z550-15 sample is lower than that of the Natural zeolite (Zini) (Table 4).

Moderate editing of the English language required

Reviewer 2 Report

The reviewed manuscript addresses an important environmental problem, organic dye removal from wastewater. The problem is tackled by using a calcined zeolite as a photocatalyst. The calcination is conducted by controlled laser irradiation. The obtained material is thoroughly characterized by modern methods of surface analysis. The manuscript is well organized and well written. The presentation is succinct and logical.

There is one issue which, on my opinion, must be addressed before this manuscript is ready for publication. Namely, the authors did not justify the use of laser irradiation for calcination and this leaves a significant leap in logic. Any reader would ask why simple heating wouldn’t be used instead. I am taking that laser irradiation, unlike heating, can be used for a controlled period of time – but the authors never mentioned this. Furthermore, the calcination time was merely mentioned as a variable – but its influence on material’s properties was not described in the manuscript. Was it important or not? To fill this gap, the authors should fill this gap by providing the pertinent data (at least, cursory) and justifying the optimum time they selected (15 s).

Specific comments:

P. 2, line 48. Introduce the AOP acronym when first mentioned.

P. 2, lines 53 and 54. Use subscripts in the chemical formulas of sulfate and superoxide.

P. 2, line 82. Fix the inconsistent word capitalization in section titles.

P. 2, line 83. Unnecessary word capitalization (zeolite). Check throughout the manuscript.

P. 7, lines 221-222. Please explain how the Si/Al ratio could change upon irradiation. Where would some silicon go to? I don’t believe its evaporation may occur at such a low temperature.

P. 9, line 260. Change “spectrums” to “spectrum.”

P. 9, Fig. 7b. This is just a calibration curve, so I suggest deleting the last sentence of the first paragraph of section 3.6.  

Good idiomatic English, barring a few minor typos

Reviewer 3 Report

The manuscript submitted for review concerns the use of a natural material - zeolite - for the degradation process of organic pollutants, using the example of the dye LY4G. At this stage I find the work interesting however incomplete. I have presented my questions and suggestions below.

1. Fig. 2 is unnecessary and the description in the text is sufficient.

2. Please indicate in what volume of water the dye used in the photodecomposition studies was dissolved.

3. Fig.4 does not show the differences described in the text.

4. Fig.5 does not show the differences described in the text.

5. In Tabs. 2 and 3, please specify the measurement uncertainty. Without this, comparison of the obtained values is impossible.

6. What is the reason for the shift in the absorption maximum of LY4G toward lower wavelengths as the dye concentration increases?

7. Why the degradation efficiency of Z55015 is the highest? Please expand the description of Fig.8 with such an analysis.

8. The biggest problem is the random analysis of materials. I wanted to ask what key the authors used to select materials for each experimental technique?

 - BET - Zini and samples calcined for 15 s,

 - SEM, EDX, XRD, adsorption - Zini and ZL55015,

 - XPS and photodegradation - Zini and calcinated at 550oC for different time.

Such selective characterization of the samples does not allow analysis of their morphological, structural, adsorption properties and activity toward photodegradation of the selected dye. The authors write 'According to the literature data, a temperature of 550°C is defined as the optimal temperature for annealing natural zeolites.' Since this is the case, I suggest focusing on the full characterization of just these materials calcined at 550oC. I also suggest that the remaining results be included in the supplementary materials.

At this stage, the work requires workload and completion of analysis. It cannot be recommended for publication.

Round 2

Reviewer 1 Report

In the revised manuscript, some comments raised in the previous review have been modified/ explained. However, in order to be published in the Journal of Materials, this manuscript is in need of major revision. The following comments can be refereed:

-          For comment 1: In the introduction, several methods for the removal of Lanasol Yellow 4G from aqueous solutions have been added.

-       For comment 2: The relationship between zeolite treatment modes, the properties and adsorption capacity as well as the photo-Fenton activity of the materials have not been added and explained.

- For comment 3: Experimental methods have not been fully described, and some information has not been added at the request of the reviewer such as lamp intensity, and thickness of the water layer.

-  For comment 4, the crystallinity degree and the crystal size of the samples have been added and listed in Table 2. However, it is advisable to comment on the relationship between the results in Tables 2 and 1, the relationship between the surface properties of zeolite and its crystallization, and how these depend on the calcination conditions.

-  Comment 5:  “Please explain why after Laser Radiation treatment pores radius of zeolite decreases; especially, sample ZL700-15 has a smaller pore radius, and larger pore volume, but the surface area (BET) is smaller than the sample treated at 300 oC and 550 oC? (Table 1.). And why “On the other hand, the calcination process carried out at 300°C and 700°C leads to a decrease in surface area in contrast to the original Zini sample.” (lines 177-178). Please explain why zeolite is not treated at 400, 450 oC, and 600 and 650 oC?” has not been resolved.

-  Comment 6 “Please explain why “For both samples, the admissible capacity as the pH increases, with the lowest capacity observed at the neutral pH of 7” and why the adsorption capacity (Q) of ZL550-15 is less than the Zini sample “ has not been resolved.

-          The remaining comments have been revised and added.

In general, the scientific explanations of the results have not been given enough attention; the manuscript is presented in the form of a report of the results. Therefore, the scientific significance has not been improved.

Moderate editing of the English language is required.

Reviewer 3 Report

The manuscript has been revised and my questions have been answered. I recommend this article for publication in Materials.

Author Response

Thank you for reviewing the revised manuscript.

We are glad to hear that your questions have been addressed and that you recommend our article for publication in Materials.

We appreciate your positive feedback and support